# New Insight into Rubber Composites Based on Graphene Nanoplatelets, Electrolyte Iron Particles, and Their Hybrid for Stretchable Magnetic Materials

**DOI:** 10.3390/polym14224826

**Published:** 2022-11-09

**Authors:** Vineet Kumar, Md Najib Alam, Sang-Shin Park, Dong-Joo Lee

**Affiliations:** School of Mechanical Engineering, Yeungnam University, 280 Daehak-ro, Gyeongsan 38541, Korea

**Keywords:** mechanical stretchability, silicone rubber, graphene nanoplatelets, electrolyte iron particles, compressive modulus, anisotropy

## Abstract

New and soft composites with good mechanical stretchability are constantly addressed in the literature due to their use in various industrial applications such as soft robotics. The stretchable magnetic materials presented in this work show a promising magnetic effect of up to 28% and improved magnetic sensitivity. The composites are soft in nature and possess hardness below 65. These composites were prepared by mixing silicone rubber with fillers such as graphene nanoplatelets (GNP), electrolyte-iron particles (EIP), and their hybrid via solution mixing. The final composites were cured at room temperature for 24 h and their isotropic and anisotropic properties were studied and presented. The mechanical properties under compressive and tensile strain were studied in detail. The results show that the compressive modulus was 1.73 MPa (control) and increased to 3.7 MPa (GNP) at 15 per hundred parts of rubber (phr), 3.2 MPa (EIP), and 4.3 MPa (hybrid) at 80 phr. Similarly, the mechanical stretchability was 112% (control) and increased to 186% (GNP) at 15 phr, 134% (EIP), and 136% (hybrid) at 60 phr. Thus, GNP emerges as a superior reinforcing filler with high stiffness, a high compressive modulus, and high mechanical stretchability. However, the GNP did not show mechanical sensitivity under a magnetic field. Therefore, the hybrids containing GNP and EIP were considered and an improved mechanical performance with magnetic sensitivity was noticed and reported. The mechanism involves the orientation of EIP under a magnetic field causing a magnetic effect, which is 28% for EIP and 5% for hybrid.

## 1. Introduction

Stretchable magnetic materials (SMM) consist of composites reinforced with different types of iron particles and polymers with a mainly elastomeric matrix [1]. SMM have a stretchable behavior when strained [2]. The mechanical properties of SMM are influenced by the type of strain [3]. The type of strain can be tensile or compressive in nature [4]. The alternative way of affecting their mechanical properties is through the influence of a magnetic field [5]. The iron particles present in SMM tend to orient in the direction of the magnetic field, thereby influencing the mechanical properties of SMM [6].

Such an orientation of the iron particles is also called an anisotropic effect, while the samples without such an orientation are called isotropic samples [7] (Figure 1). However, (a) the influence of the mechanical properties orienting the iron particles or (b) the addition of iron particles as a source of reinforcement constitute insufficient pathways for obtaining devices with industrial value [8]. So, other reinforcing fillers must be added to achieve optimum mechanical properties [9]. These fillers can be carbon black [10], carbon nanotubes [11], or graphene [12]. Among them, carbon black is traditionally used as a filler that improves mechanical properties but is used at a high content [13]. This high content alters the viscoelastic properties of the samples [13]. Thus, nanofillers such as carbon nanotubes or graphene are employed to improve and obtain the desired mechanical properties at a low filler content [14,15].

The elastomeric matrix family is quite large, including rubber as one member [16]. The rubber matrix can be synthetic [17] or natural in origin [18]. Rubber with a natural origin is known as “natural rubber latex”, and is obtained from trees [19]. Synthetic rubber is dynamic, with vast classes ranging from diene rubber [20] to silicone rubber [21]. Among them, silicone rubber is more promising than diene rubber in terms of hardness, easy processing, and easy curing [22]. Silicone rubber is categorized based on the type of vulcanization, which can be room temperature or high temperature [23]. Among them, room temperature-vulcanized silicone rubber is more promising due to its versatile behavior, softness, and curability without the use of sophisticated machines [24].

Nanofillers based on carbon allotropes added to silicone rubber lead to drastic improvements in the composites’ mechanical, electrical, or thermal properties [25]. A review study by Kumar et al. showed that these improved properties may be useful for a range of soft industrial applications such as strain sensors [26]. The review study by Kumar et al. showed that among the different ranges of carbon-based nanofillers, CNT and graphene emerge as the best candidates for reinforcement [26]. These improved properties are due to (a) the high aspect ratio of these nanofillers [27]; (b) the favorable morphology of these nanofillers, which allows for their uniform dispersion [28]; and (c) the high interfacial area of these nanofillers, which enables high stress-transfer from polymers to these nanofillers in composites [29].

Moreover, a silicone rubber matrix filled with iron particles may be useful for “magneto-rheological elastomers” [30] or SMM, in which mechanical properties can be influenced by switching magnetic fields, as performed in this work [1,30]. The mechanism behind such an increase is the orientation of the iron particles in the direction of a magnetic field, thereby forming a chain-like structure and influencing the mechanical properties such as the modulus [31]. Another way to improve these mechanical properties is to add a reinforcing filler along with iron particles to obtain high-performance SMM [32,33]. Thus, graphene nanoplatelets were used in the present work along with electrolyte iron particles in a silicone rubber matrix.

Various studies have been reported that show the use of binary carbon-based reinforcing fillers such as carbon nanotubes [34], graphene [35], or carbon black [36] along with different types of iron particles in the rubber matrix to obtain SMM [1,34,35,36]. These studies show that the incorporation of secondary fillers based on carbon not only improves mechanical performance but also does not influence the magnetic sensitivity exhibited by the iron particles [37]. This study is an advancement from the previously reported studies because it studies the anisotropic effects of fillers at a high magnetic field of 1 Tesla, wherein the effect of the orientation of EIP on the dispersion of GNPs was correlated. Moreover, hybrid composites were also prepared in this study that investigates the composites’ possible synergistic effects and their relation to the improvement of various mechanical properties. We hypothesize that the hybrid composite possesses the advantages of a higher compressive modulus, increased reinforcing effects, and optimum magnetic sensitivity, while EIP possesses the advantages of an increased magnetic effect, which was supported experimentally in this work.

## 2. Materials and Methods

### 2.1. Materials

The RTV-silicone rubber was used as a rubber matrix in the present work and was purchased from Shin-Etsu Chemical Corporation Limited, Tokyo, Japan. Its commercial name is “KE-441-KT”, and it is transparent in nature. The vulcanizing agent used in the present work was “CAT-RM”, which was purchased from Shin-Etsu Chemical Corporation Limited, Tokyo, Japan. The graphene nanoplatelets were used as a reinforcing nanofiller in the present work. Their commercial name is “XG C750” and they were purchased from XG Science, Lansing, MI, USA. The nanoplatelets had a total surface area of around 750 m^2^/g, lateral dimensions from 500 nm–1 µm, and thickness of 1–2 nm. The micron-size electrolyte iron particles (EIP) with the commercial name “Fe#400” were obtained from Aometal Corporation Limited, Gomin-si, Korea. The average particle size of each EIP is >10 µm, with each particle possessing an irregular shape, being light greyish in color, possessing a density of 2–3 g cm^−3^, and a purity of 98.8% iron, while other traces of carbon, oxygen, and nitrogen were also found. All the materials were used in a pristine state without any further purification. The mold-releasing agent was purchased from Nabakem, Pyeongtaek-si, Korea.

### 2.2. Fabrication of Rubber Composites

The steps of composites’ preparation were optimized and reported in previous studies [38]. The procedure involved spraying the molds with mold-releasing agents and then drying them at room temperature for 3 h. In the next step, the liquid RTV-SR rubber was poured into a beaker and a known amount of filler (Table 1) was mixed in. Rubber–filler mixing was performed for around 10 min. Next, the known amount of vulcanizing agent was added to the sample and mixed for nearly 1 min. Then, the composite was added to the molds and kept for 24 h at ambient conditions before the vulcanized composite was ready (Figure 2) for testing of mechanical and anisotropic properties.

### 2.3. Characterization Technique

The nanofiller’s morphology and its dispersion in the rubber matrix were studied by SEM (S-4800, Hitachi, Tokyo, Japan). The composite specimen was sectioned to a thickness of 0.5 mm using a surgical blade and then placed on the SEM stub before the coating process was initiated. The SEM samples were coated with conductive platinum for 2 min to make the surface of the samples electrically conductive. The mechanical properties under compressive and tensile strain were studied using a universal testing machine (UTS, Lloyd Instruments, Bognor Regis, UK). The mechanical properties under compressive strain were measured at a 4 mm/min strain rate and under a load of 0.5 kN using cylindrical samples. These cylindrical samples were 10 mm in thickness and 20 mm in diameter. The maximum strain of 35% was applied to these cylindrical samples as higher strain leads to fracture of the sample. The tests of the mechanical properties under tensile strain were performed at a strain rate of 100 mm/min on a dumbbell-shaped sample with a gauge length of 25 mm and thickness of 2 mm. The tensile specimens were strained until fracture and their mechanical parameters such as their moduli, tensile strength, or fracture strain were estimated. These mechanical properties were obtained following DIN 53 504 standards. The anisotropic magnetic properties were studied at 1 T by placing the specimen inside a magnetic field for 90 min.

## 3. Results and Discussion

### 3.1. Morphologies of Nanofillers

The morphology of the nanofillers used as a reinforcing agent in the rubber matrix is known to affect the properties of composites [39]. Therefore, the morphology of the fillers was studied and is presented in Figure 1. Figure 1a shows the typical platelet-like morphology of GNPs. The platelets in the graphene nanoparticles were three-dimensional in nature [40]. The particle size in the lateral dimension was in the range from 500 nm–1 µm, and the thickness of 1–2 nm led to a very high aspect ratio. This high aspect ratio provides very high mechanical, electrical, and thermal properties when added as filler in the composite [41,42]. It is also expected that the high aspect ratio of GNPs leads to the formation of long-range and connective filler networks throughout the rubber matrix [43]. These filler networks with filler–filler and polymer–filler interactions within the composite significantly improve mechanical properties [44]. GNPs have a strong lubricating effect and are known to improve the fracture strain of the composites [45]. The GNPs’ structure includes the 3-D arrangement of 2-D graphene sheets held together by weak Vander Waals forces. On the other hand, EIP are micron-sized particles—namely, with sizes in the range of 10–12 µm—possessing a 3-dimensional morphology. These particles are magnetically active and algin themselves in the presence of a magnetic field in the composite. This behavior of these iron particles makes them promising in terms of their anisotropic effects and potential magnetic sensitivity applications [46]. The micron-sized EIP are rough, have an irregular shape, and can be easily aligned under a magnetic field. This anisotropic property of EIP leads to an improvement in the mechanical properties of the composites and will be discussed in the coming sections.

### 3.2. Filler Dispersion Analyzed through SEM Microscopy

The effect of the filler dispersion on the properties of the composites is well-known. It is also known that composites with a uniform filler dispersion exhibit more optimum properties than those with aggregated fillers or non-uniform dispersion. Thus, the study of the filler dispersion in composites is a key aspect for ascertaining their properties. In this study, the filler dispersion was studied through SEM images. A number of images were studied, and their representative images are presented in Figure 2. Figure 2a–c show SEM images of the control sample. It is evident that there are no filler particles, as expected. Then, different types of fillers such as GNP, EIP, or their hybrids were added, and their dispersion was studied. From Figure 2d–f, it was found that the GNP particles are dispersed uniformly while very few aggregates can be noticed at a high resolution, as in Figure 2e. However, the influence of these aggregates is not severe enough to affect the composites’ properties. So, the properties of the GNP-filled composites were expected to be higher, and were studied, as shown in Figure 3, Figure 4, Figure 5 and Figure 6. Figure 2g–i shows the dispersion of EIP in the rubber matrix. It can be observed from the SEM images that the EIP are also uniformly distributed. However, due to the large particle size of the EIP, the surface roughness was higher and there were fewer EIP when compared to the GNPs’ particle distribution in the composites. Similarly, the distribution of the hybrid filler was studied in Figure 2j–l. It was found that, in general, the GNP particles are found in the vicinity of the EIP, and the interfacial interaction of the EIP in the composites could be improved by the GNP particles. So, a sort of synergistic aspect was generated in their dispersion and this led to better properties in the hybrid composites.

### 3.3. Mechanical Properties under Compressive Strain

Composites’ mechanical properties depend on the distribution of the filler [47]; the filler’s characteristics [48], such as the type of filler, the nature of the filler, the shape of the filler, the size of the filler, or the filler’s aspect ratio; the cross-linking density of the curatives [49]; the type of rubber matrix [50]; and the type of applied strain under which the mechanical properties are tested [51]. Herein, fillers with different characteristics were used in silicone rubber and their effects on the rubber’s mechanical properties were tested. Figure 3a–c provide the compressive stress–strain behavior of the composites with different types of fillers and their hybrids. All the stress–strain profiles show that the stress increases with an increase in the compressive strain. This trend is attributed to an increase in the packing fraction of the polymer chains and filler particles with the increasing compressive strain [51]. This process increases the stiffness in the composite, thereby leading to higher compressive stress at a higher level of compressive strain. It is also interesting to note that the GNP-filled composites show higher compressive strain values at all strain levels, even at lower filler loadings. These results agree with the results obtained for GNPs in the literature [52]. This phenomenon is due to the higher aspect ratio of the GNPs, which induces improved filler networking of the GNPs in the composites leading to higher compressive stress [52]. It is also interesting to note that the hybrid composite shows improved stress resistance, which could be due to the higher filler loading and synergism among the binary filler particles in the composite [53].

The behavior of the compressive modulus at different filler loadings is presented in Figure 3d. The GNPs show excellent modulus values—even at a lower loading—than the EIP. This is attributed to (a) the high aspect ratio of the GNPs, which led to the formation of continuous filler networks throughout the rubber matrix even at a lower filler loading [52]; (b) the high surface area of the GNPs, which leads to the availability of a higher interfacial area that allows for better stress transfer from the polymer to the filler particles [54]; and (c) the high levels of filler–filler and polymer–filler interactions due to the presence of the large interfacial area and small particle size of the GNPs [55]. Moreover, the hybrid specimen shows a significantly higher compressive modulus than the EIP at the same filler loading. This can be attributed to (a) the higher reinforcing ability of the GNPs in the hybrid filler that leads to a remarkable increase in the compressive modulus, and (b) the favorable positive synergism among the binary fillers that leads to a remarkable increase in the compressive modulus [53]. In the end, the EIP show poor reinforcing properties in all the composites. This is attributed to (a) the micron-sized particles that lead to poor stress transfer even at high filler loadings and their small aspect ratio that leads to the formation of non-efficient and discontinuous filler networks.

### 3.4. Mechanical Properties under Tensile Strain

The stress–strain behavior of the different composites was studied and presented in Figure 4a–c. For all the composites, it was witnessed that the stress increases with the increasing strain until fracture. This behavior was attributed to improved interfacial interaction between the filler and rubber matrix [56]. The increase in tensile strain leads to filler particles and polymer chains oriented against the direction of the applied strain [57]. This causes mechanical resistance against the applied strain and leads to an increase in stress with the increasing strain [58].

It is also interesting to note that the GNP-based composites show higher tensile stress and fracture strain at the applied strain than EIP and the hybrid composites. The improved stress is due to the efficient filler networking that allows for better stress transfer within the composites [59]. The improved fracture strain is due to the lubricating effect of the GNPs, which is higher with a higher number of GNPs [60].

The behavior of other mechanical properties such as the moduli (Figure 4d), tensile strength (Figure 4e), and fracture strain (Figure 4f) are presented. Among all the experimental data, the control sample shows the lowest mechanical strength and stiffness. This is due to the absence of reinforcing fillers in virgin rubber. Besides this, it was found that GNP exhibited outstanding performance among all the fillers studied irrespective of the smaller loadings, showing values of up to 15 phr greater than EIP and their hybrid at 80 phr. The tensile modulus was higher for the GNP-filled composites followed by the hybrid composites and was lowest for the EIP-filled composites. This result is attributed to the high reinforcing effect of the GNPs due to their favorable morphological features such as a high aspect ratio [61].

The EIP show poor properties due to their poor reinforcing effects resulting from their poor morphological features, such as their large particle size and small aspect ratio. The hybrid specimen shows medium modulus values, which are higher than the EIP and lower than the GNPs. This is due to the semi-effect of the GNPs in the hybrid composite that forms synergism among the GNPs and EIP in the hybrid specimen. Moreover, as expected, the tensile strength and fracture strain were also superior for the GNP-based filler. This behavior agrees with the mechanical performance obtained in Figure 3 and Figure 4a–d. The higher tensile strength is due to the improved filler network formation of the GNPs, its interaction with the rubber matrix, and its ability to achieve good stress transfer from the rubber matrix to GNP particles [62]. The improved fracture strain of the GNPs is due to their lubricating effect and favorable platelet morphology that form three-dimensional networks in the rubber matrix, leading to higher fracture strain.

### 3.5. Theoretical Models and Hardness of Composites

The prediction of mechanical properties is a well-known subject of research in the literature [63]. These models help to estimate the deviation of experimental results from theoretical models. The existing theoretical models such as the Guth–Gold Smallwood equations [64,65] were used in the present work to understand their deviation from the experimental findings. These models are known to depend on the aspect ratio of fillers such as GNP or EIP for the one component and the hybrid for the two-component system, the filler volume fraction of the fillers, and their interactive factors, especially for the hybrid two-component system. The Guth–Gold Smallwood equations for a one and two-component system [66] are as follows:E_GNP_ = E_o_ (1 + 0.67f_GNP_ϕ_GNP_)(1)
E_EIP_ = E_o_ (1 + 0.67f_EIP_ϕ_EIP_)(2)
E_GNP+EIP_ = E_o_ [(1 + 0.67f_GNP_ϕ_GNP_) + (1 + 0.67f_EIP_ϕ_EIP_)] × a(3)
where E_o_ is the modulus of the control composite, E_GNP_ is the predicted modulus for GNP, E_EIP_ is the predicted modulus for EIP, E_GNP+EIP_ is the predicted modulus for the hybrid filler, f_GNP_ is the aspect ratio of GNP, f_EIP_ is the aspect ratio of EIP, ϕ_GNP_ is the volume fraction of GNP, ϕ_EIP_ is the volume fraction of EIP, and a is the interacting factor of the binary fillers in the composite.

From Figure 5a,b, it is evident that our experimental data are in good agreement with the theoretical models, except for the hybrid data of the tensile modulus in Figure 4b. The agreement of the experiments with the theoretical model confirms that the experimental data are reliable and is an important contribution to the composite field. However, the deviation of the hybrid modulus with the predicted modulus in Figure 5b after 60 phr could be due to the aggregation of the binary fillers that leads to a decrease in the modulus and the deviation of the results. In principle, the model assumes perfect dispersion and perfect filler–polymer interaction, which are hard to establish experimentally. So, few results deviate from the predicted model [66,67].

The hardness of the composites was studied to determine whether each filled composite was hard or soft in nature. In many cases, if the hardness of the composites falls below 65, it is termed a soft composite [68]. These soft composites can be useful for industrial applications such as soft robotics [69]. In the present work, the hardness of different composites was studied and is presented in Figure 5c. It was found that the GNP composite shows a more robust hardness increase with the addition of GNPs compared to the hybrid and EIP. The higher hardness in the GNP composite can be attributed to the higher reinforcing effect of GNPs, which is due to favorable features such as their high aspect ratio. These results agree with the mechanical data presented in Figure 3 and Figure 4 in the mechanical properties section.

### 3.6. Reinforcement by Particulate Nanofillers

The reinforcing factor and reinforcing efficiency of the different composites under compressive and tensile strain were studied and are presented in Figure 6. The reinforcing factor was calculated by
(4)$$R.F.=\frac{\mathrm{EF}}{\mathrm{Eo}}$$
where R.F. is the reinforcing factor, EF is the modulus of the filled composites, and Eo is the modulus of the control specimens. Similarly, the reinforcing efficiency [70] was calculated:(5)$$R.E.\;\mathrm{at}\;\mathrm{compressive}\;\mathrm{strain}=\frac{\sigma \;\left(35\%\right)\mathrm{filled}-\sigma \;\left(35\%\right)\mathrm{unfilled}\;}{\mathrm{wt}\%\;\mathrm{of}\;\mathrm{filler}}$$
(6)$$R.E.\;\mathrm{at}\;\mathrm{tensile}\;\mathrm{strain}=\frac{\sigma \;\left(100\%\right)\mathrm{filled}-\sigma \;\left(100\%\right)\mathrm{unfilled}\;}{\mathrm{wt}\%\;\mathrm{of}\;\mathrm{filler}}$$where R.E. is the reinforcing efficiency; σ is the stress at a particular strain, which is 35% for compressive strain and 100% for tensile strain; and wt% is the weight of the filler. The compressive and tensile R.F. (Figure 6a,b) show that the GNPs have a significant reinforcing effect over the EIP-filled composites. This can be attributed to the small particle size of the GNPs that disperse in the rubber matrix in a highly exfoliated state [71].

Besides this, the EIP have a poor reinforcing effect due to their large particle size, which is dispersed unevenly in the rubber matrix. The filler networks are also more efficient in the GNP-filled composites than in the EIP-reinforced ones. These results agree with the mechanical properties studied in Figure 3, Figure 4 and Figure 5. Moreover, the compressive and tensile R.E. (Figure 6c,d) also show a superior reinforcing effect of the GNPs in the rubber matrix. These results agree with the results presented in the mechanical properties section in Figure 3, Figure 4 and Figure 5. It is also interesting to note that the R.E. decreased with an increase in the filler loading. This trend can be explained by Equations (5) and (6), wherein the R.E. is inversely proportional to the filler loading [70]. So, when the filler loading increases, the R.E. decreases for all the filled composites. It is noteworthy that the hybrid shows higher R.E. than the EIP-filled composites at all filler loadings. This can be attributed to the synergism between the binary fillers in the hybrid composite. The EIP-filled composites show poor R.E. at all loadings due to their poor aspect ratio and large particle size that is unevenly distributed in the rubber matrix even at large filler loadings of 60–80 phr.

### 3.7. Anisotropic Magnetic Effect in Mechanical Properties

The iso-anisotropic stress–strain curves are presented for the EIP-filled composites (Figure 7a) and hybrid composites (Figure 7b). Two types of effects were noticed from the stress–strain profiles: (a) a stress change due to the effect of the magnetic field in the composites, and (b) a stress change due to a change in the compressive strain [72,73]. The first change in stress under a magnetic field is due to the orientation of EIP in the direction of the magnetic field, which leads to a change in compressive stress. The second change in stress under compressive strain is due to the packing of filler particles, which increases with the increases in stress, as described previously in Figure 3.

Figure 7c shows the behavior of the compressive modulus in the presence and absence of a magnetic field. It can be seen that when the composites containing EIP or the hybrid were subjected to a magnetic field, the compressive modulus increased. This increase in the compressive modulus could be due to the orientation of the iron particles forming chain-like structures in both the EIP and hybrid composites [74]. Such an effect is also called the anisotropic effect, as described in Figure 7d. The anisotropic effect is defined as an effect in which a change in mechanical stiffness occurs when isotropic composites are subjected to a magnetic field [75].

It is interesting to note that the anisotropic effect was higher for the EIP-filled composites than for the hybrid composites, even though the modulus was higher in the hybrid composites. The higher anisotropic effect in the EIP is due to the greater number of vacancies in the composite containing EIP, which allows them to orient freely thereby causing a higher degree of uniform orientation. On the other hand, the GNPs in the hybrid occupy almost all the vacancies, thereby making it difficult for the EIP to align. Due to this reason, the anisotropy was higher in the EIP than in the hybrid composites.

### 3.8. Mechanism of the Anisotropic Magnetic Effect

Different types of filler–polymer microstructures are formed in the presence and absence of a magnetic field. In the present work, the magnetic field was applied in the pre-curing state of the composite. The magnitude of the magnetic field was 1 T, and the duration of the field was 90 min. It was found from the tests that the EIP are oriented in the direction of the magnetic field, thereby influencing their mechanical properties such as their moduli, as presented in Figure 7.

The mechanism involves the presence of a restorative force that opposes the field-aligned orientation of the magnetic particles in the composites. Therefore, the magnetic field should be higher than this restorative force to orient the magnetic EIP. It is clear from Figure 8 that the restorative force in the case of the hybrid filler is higher than in the EIP-filled composites. So, under the same magnetic field, the degree of orientation of the EIP is lower in the hybrid composites than in the EIP composites.

There are three stages of the microstructure formation of the filler and polymer chains in the composites during the curing process. These processes are (a) the isotropic stage, in which the filler particles are randomly distributed in all the composites investigated (0 T of magnetic field); (b) the intermediate stage, in which the EIP or magnetic particles are partially oriented at 1 T of the magnetic field; and (c) the mature or anisotropic stage, in which the EIP orientation attains maturity and the magnetic exposure of 90 min—as in the present work—is achieved. The isotropic and anisotropic effects on the mechanical properties are presented in Figure 7. The alignment of the EIP under a magnetic field could be due to several reasons. However, the key factors that influence their orientation could be (a) the morphology, shape, size, and polymer–filler interactions in the composite; (b) the occurrence of a field-induced dipole among the magnetic filler particles [76]; or (c) the filler networking density, which significantly affects the orientation of the magnetic filler particles. These features of MREs are useful for various applications such as electromagnetic absorbers and various other functions as reported in the literature [77,78,79,80].

## 4. Conclusions

The robust performance of the stretchable magnetic materials is presented in this work. After the successful preparation of the composites by solution mixing, the mechanical properties of the composites were studied in the presence and absence of a magnetic field. The mechanical properties were investigated under compressive and tensile strain. The results obtained from these mechanical properties show that both the compressive modulus and mechanical stretchability increase with an increasing filler content, wherein GNPs emerge as an outstanding filler material. For example, the compressive modulus was 1.73 MPa (control) and increased to 3.7 MPa (GNP) at 15 per hundred parts of rubber (phr), 3.2 MPa (EIP), and 4.3 MPa (hybrid) at 80 phr. Similarly, the mechanical stretchability was 112% (control) and increased to 186% (GNP) at 15 phr, 134% (EIP), and 136% (hybrid) at 60 phr. Moreover, the GNPs show a higher reinforcing factor and reinforcing efficiency; however, the GNPs lack magnetic sensitivity due to a lack of iron particles and are not suitable for stretchable magnetic materials. Therefore, EIP and a GNP hybrid were prepared and studied. The mechanism involves the orientation of EIP under a magnetic field causing a magnetic effect, which is 28% for EIP and 5% for the hybrid. Thus, EIP and the hybrid-filled composites emerge as promising candidates for stretchable magnetic materials, which is the main theme of this work.

## Data Availability

Not applicable.

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
