# Peer review of "New Insight into Rubber Composites Based on Graphene Nanoplatelets, Electrolyte Iron Particles, and Their Hybrid for Stretchable Magnetic Materials"

_polymers, 2022, doi:10.3390/polym14224826_

Round 1
Reviewer 1 Report
The authors prepared soft nanocomposites by mixing silicone rubber with fillers such as graphene nanoplatelets (GNP), electrolyte-iron particles (EIP), and their hybrid via solution mixing. The morphology, mechanical properties and anisotropic magnetic effect in mechanical properties were characterized by SEM, compressive and tensile tests. However, this article is written like a scientific report.
1. The scheme 3 can be removed.
2. “From Figure 2d-2f, it was found that the GNP particles are dispersed uniformly while very few aggregates can be noticed at high resolution as in Figure 2f.” It cannot clearly see GNP particles from SEM images in Figure 2 (d-f).
3. The author should check again for some spelling errors and mistakes.
Author Response
Responses to the Editor’s and Reviewers’ comments on the Manuscript (Polymers-1985272-R1)
We have now revised our manuscript based on Editor’s and Reviewers’ valuable comments and suggestions. All changes and corrections are highlighted in RED color in the manuscript. Here, we hope that the Editor and Reviewers will be satisfied with our responses and thereby our revised manuscript. We are much thankful to the Editors and Reviewers for giving us a chance to improve the manuscript for publication in Polymers Journal.
Reviewer #1
The authors prepared soft nanocomposites by mixing silicone rubber with fillers such as graphene nanoplatelets (GNP), electrolyte-iron particles (EIP), and their hybrid via solution mixing. The morphology, mechanical properties, and anisotropic magnetic effect in mechanical properties were characterized by SEM, compressive and tensile tests. However, this article is written like a scientific report.
Response: Thank you for your kind comment. The paper is improved as desired.
1) The scheme 3 can be removed.
Response: Thank you for the suggestion. Scheme-3 is removed as desired.
2) “From Figure 2d-2f, it was found that the GNP particles are dispersed uniformly while very few aggregates can be noticed at high resolution as in Figure 2f.” It cannot clearly see GNP particles from SEM images in Figure 2 (d-f).
Response: Thank you for the suggestions. The high resolution of isolated GNP particles is provided in Figure 2f as desired by the reviewer.
3) The author should check again for some spelling errors and mistakes.
Response: Thanks for the suggestion. The paper is proof-read again for errors and mistakes as desired.
We again like to thank the reviewer for their critical suggestions and for improving the manuscript.

Reviewer 2 Report
This manuscript reports a new insight on rubber composites based on graphene nano platelets, electrolyte iron particles. The composites are supposed as stretchable magnetic materials for multiple applications. This manuscript is well organized and the data support the conclusions very well. However, minor revision is requested before its acceptance. My comments are as below:
1. The size and the component information of Fe#400 should be given, along with other information as much as possible.
2. Does the irregular morphology of EIP affect the final properties of the composite?
3. Higher resolution SEM images or EDS mapping images should be given in Figure 2 to distinguish EIP from GNPs.
4. Dielectric and magnetic properties of composite materials are required.
5. The authors should explore the potential applications of these composites to enhance the research background of this manuscript. Some references on electromagnetic research field are suggested such as Nano-Micro Lett,2022,4(1):11; J Mater Sci Technol,2022,113(20): 33-39; Chem. Eng. J.,2022,430(4): 133178; Compos. B. Eng.,2021,226: 109335; et al.
Author Response
Responses to the Editor’s and Reviewers’ comments on the Manuscript (Polymers-1985272-R1)
We have now revised our manuscript based on Editor’s and Reviewers’ valuable comments and suggestions. All changes and corrections are highlighted in RED color in the manuscript. Here, we hope that the Editor and Reviewers will be satisfied with our responses and thereby our revised manuscript. We are much thankful to the Editors and Reviewers for giving us a chance to improve the manuscript for publication in Polymers Journal.
Reviewer #2
This manuscript reports a new insight on rubber composites based on graphene nano platelets, electrolyte iron particles. The composites are supposed as stretchable magnetic materials for multiple applications. This manuscript is well organized and the data support the conclusions very well. However, minor revision is requested before its acceptance. My comments are as below:
Response: Thank you for the comment and suggestion.
(1) The size and the component information of Fe#400 should be given, along with other information as much as possible.
Response: Thank you for the comment. The average particle size of EIP is > 10 µm, the shape is irregular, its color is light greyish, its density is 2-3 g cm-3, its purity is 98.8% iron, and other traces of carbon, oxygen, nitrogen is found.
(2) Does the irregular morphology of EIP affect the final properties of the composite?
Response: Thank you for the comment. The morphology and particle size of the particles are well-known to affect the final magnetic and mechanical properties. For example, carbonyl-iron particles with oval morphology can easily orient under a magnetic field leading to a higher magnetic field than the irregular morphology of EIP under the same filler loading.
(3) Higher resolution SEM images or EDS mapping images should be given in Figure 2 to distinguish EIP from GNPs.
Response: Thank you for the comment. The high resolution of an isolated graphene sheet is provided in Figure 2f. For other high-resolution images, since the particle size of EIP is >10 µm. So, the provided SEMs are the best resolution to study the micron-size particles. EDX is not useful because carbon is present in both silicone rubber and graphene. So, it’s hard to distinguish between them and this makes this technique non-useful for such composites.
(4) Dielectric and magnetic properties of composite materials are required.
Response: Thank you for the comment. Sorry, we don’t have the facility to perform dielectric properties. For magnetic properties, please refer to Figure 7 and the mechanism of magnetic effect in Figure 8. The mechanism involves the orientation of EIP under a magnetic field causing a magnetic effect which is 28% for EIP and 5% for hybrid. Further studies on magnetic sensitivity such as magnetic response rate and stress relaxation properties under a magnetic field will be studied in future manuscripts.
(5) The authors should explore the potential applications of these composites to enhance the research background of this manuscript. Some references on electromagnetic research field are suggested such as Nano-Micro Lett,2022,4(1):11; J Mater Sci Technol,2022,113(20): 33-39; Chem. Eng. J.,2022,430(4): 133178; Compos. B. Eng.,2021,226: 109335; et al.
Response: Thank you for the suggested references. All the references are cited as desired. Please refer to the reference section for new cited text.
We again like to thank the reviewer for their critical suggestions and for improving the manuscript.

Reviewer 3 Report
The manuscript deals with the development of strechtable magnetic materials, what corresponds with general scientific trend. The manuscript is well addressed and structured. The results are clearly presented and accompanied by a valuable discussion. I have only one suggestion - "virgin" should be replaced by "control" or "reference". After optional minor revision I recommend to accept.
Author Response
Responses to the Editor’s and Reviewers’ comments to the Manuscript (Polymers-1985272-R1)
We have now revised our manuscript based on Editor’s and Reviewers’ valuable comments and suggestions. All changes and corrections are highlighted in RED color in the manuscript. Here, we hope that the Editor and Reviewers will be satisfied with our responses and thereby our revised manuscript. We are much thankful to the Editors and Reviewers for giving us a chance to improve the manuscript for publication in Polymers Journal.
Reviewer #3
The manuscript deals with the development of strechtable magnetic materials, what corresponds with general scientific trend. The manuscript is well addressed and structured. The results are clearly presented and accompanied by a valuable discussion. I have only one suggestion - "virgin" should be replaced by "control" or "reference". After optional minor revision I recommend to accept.
Response: Thank you for the comment and suggestion. The suggestion is done as desired. Please refer to Figure 3 and Figure 4 for correction.
We again like to thank the reviewer for their critical suggestions and for improving the manuscript.

Round 2
Reviewer 1 Report
Accept in present form.